# Differential Regulation of Nav1.1 and SCN1A Disease Mutant Sodium Current Properties by Fibroblast Growth Factor Homologous Factors

**DOI:** 10.3390/cells14040291

**Published:** 2025-02-15

**Authors:** Ashley Frazee, Agnes Zybura, Theodore R. Cummins

**Affiliations:** 1Biology Department, School of Science, Indiana University Indianapolis, Indianapolis, IN 46202, USA; amfrazee@iu.edu (A.F.);; 2Stark Neurosciences Research Institute, Indianapolis, IN 46202, USA

**Keywords:** channelopathy, voltage-gated sodium channel, migraine, inactivation, FHF2A

## Abstract

Fibroblast growth factor homologous factors (FHFs) regulate the activity of several different voltage-gated sodium channels (Na_v_s). However, more work is needed to determine how specific FHF isoforms and variants affect the properties of different Na_v_ isoforms. In addition, it is not known if FHFs can differentially modulate the properties of Na_v_ variants associated with disease. Here, we investigated the effects of FHF2A and FHF2B on Nav1.1 properties as well as on a familial hemiplegic migraine 3 (FHM3) causing mutation in this channel, F1774S. We found that FHF2A, but not 2B, induced prominent long-term inactivation (LTI) in the wild-type (WT) Nav1.1. Interestingly, FHF2A induced LTI in the F1774S FHM3 mutant channel to a greater extent than in the WT. Furthermore, persistent currents caused by the F1774S mutation were attenuated by the co-expression of FHF2A, leading to a possible rescue of the mutant channel phenotype. By contrast, the P1894L mutation, which is associated with epilepsy and mild intellectual disability, greatly attenuated the LTI induced by FHF2A. Overall, our data show for the first time that FHF2A might be a significant modulator of Nav1.1 that can differentially modulate the impact of Nav1.1 disease-associated mutations.

## 1. Introduction

Voltage-gated sodium channels (Na_v_s) are responsible for the initiation and propagation of action potentials in neurons as well as muscle cells. Although ample evidence demonstrates that Na_v_ mutations can alter biophysical properties, the extent to which this can be modified by accessory proteins is unclear. The Na_v_ channels allow sodium to pass into the cell via a pore-forming α subunit. This α subunit is composed of four homologous domains (DI-DIVs), which are each composed of six transmembrane segments (S1–S6). Segments S1–S4 form the voltage sensors, while S5–S6 form the pore. One member of the Na_v_ family, Na_v_1.1, is found throughout the nervous system, with substantial expression in both the central nervous system (CNS) inhibitory interneurons, where it plays a major role in regulating the excitability of GABAergic neurons [1,2,3], and the peripheral nervous system (PNS) excitatory neurons, where it is heavily expressed in larger diameter neurons and has been implicated in mechanical pain sensations [4,5]. Mutations to the *SCN1A* gene that encodes Nav1.1 have been implicated in various disease phenotypes, including epilepsy and familial hemiplegic migraine 3 (FHM3) [1,2,3,6,7,8,9,10]. FHM3 is an inherited form of migraine associated with hemiparesis during the aura phase.

One FHM3 mutation, F1774S, is located in the middle of the sixth transmembrane segment (S6) of the fourth domain (DIV), as shown in Figure 1. The change from phenylalanine, which contains a hydrophobic side chain, to serine, with its polar neutral side chain, can lead to changes in sodium current properties and channel gating. Past studies have shown that this mutation causes increased persistent current (I_NaP_) in HEK-293 cells transfected with the mutant channel that was attenuated by the application of GS967, a persistent sodium current blocker [2]. I_NaP_ typically is a small portion of the total sodium current that fails to fully inactivate during prolonged depolarization [11]. An increased I_NaP_ can lead to increased firing frequency and excitability [12]. Multiple FHM3 mutations have been identified as having major gain of function effects driven by increased I_NaP_ [13].

Fibroblast growth factor homologous factors (FHFs) are known regulators of Na_v_s and can modify the function of many different sodium channel isoforms [14,15]. FHFs, while part of the FGF gene family, are functionally distinct from other FGFs. This subgroup comprises four FGF genes (*FGF11-14* encoding FHF3, FHF1, FHF2 and FHF4, respectively). The FHFs encoded by these genes remain intracellularly and can bind to the c-terminal domain (CTD) of several Na_v_s [14,16]. FHFs are expressed throughout both the central and peripheral nervous systems [14]. FHFs can have multiple splice variants, and several of the longer (A-type) splice variants have been shown to induce long-term inactivation (LTI) in other sodium channel isoforms such as Nav1.2, Nav1.5, and Nav1.6 following repetitive stimulation [15,17,18,19]. The LTI induced by A-type FHFs is characterized by rapid entry during depolarizations into a non-conducting, refractory state from which recovery is much slower than the recovery from classic fast inactivation [18]. Due to its fast onset and slow recovery, LTI can result in rapid accommodation of action potential firing [20].

While much is known about how FHFs modulate the gating properties of Nav1.2, Nav1.5, Nav1.6, Nav1.7, and Nav1.8, it is unclear if FHFs alter Nav1.1 current properties. Some evidence suggests that FHF1 does not bind to the C-terminus of Nav1.1 channels [16], but the highly conserved cores of FHF2A (also referred to as FGF13S), FHF3A, and FHF4B were shown to bind the C-terminal domain (CTD) of Na_v_1.1 using a biosensor chip analysis [21]. Wang et al. [16] reported that the interaction of the FHFs with VGSCs depends on a portion of the CTD between amino acids 1773 and 1908 of the channel. They found the binding site on the FHF to be in the core domain and specifically to involve a proline at position 149 (P149) in FHF1B (FGF12B), corresponding to P154 in FHF2B (also referred to as FGF13U) [16]. We focused on FHF2 as prior studies have shown that FHF2 is found in both trigeminal ganglia neurons as well as inhibitory interneurons, although it is not yet known which FHF2 splice variants are expressed in these locations [22,23]. Several studies have indicated that Nav1.1 currents in CNS interneurons play an important role in migraines [24,25]. However, the peripheral administration of CGRP has been shown sufficient to trigger migraine attacks [26], suggesting the importance of the excitatory peripheral neurons in this disease as well. FHF2 mutations have been identified in individuals with epilepsy where the change to the FHF protein results in a reduction in LTI, likely causing gain of function effects on overall circuit excitability, leading to infantile onset of developmental delay and intractable focal seizures [27]. These data suggest that the FHF2A modulation of voltage-gated sodium channels in both the CNS and the PNS might be of considerable importance.

Here, we investigated if FHFs could modulate Nav1.1 properties as well as Nav1.1-F1774S channels. To further interrogate the interaction between FHF2 isoforms and Nav1.1, we also examined another mutation in Nav1.1, P1894L, which is associated with epilepsy and mild intellectual disability [28]. The P1894L mutation is located in the CTD of the Na_v_1.1 channel within the region where the FHF core binds (Figure 1). Prolines are normally found on the surface of a protein and can aid in folding, while leucines have a hydrophobic side chain. The change from proline to leucine could lead to the altered or inhibited binding of FHF regulatory proteins as well as changes to channel inactivation.

We investigated the differential effects of FHF2A and FHF2B co-expression on Na_v_1.1 currents with and without the F1774S or the P1894L mutation. We hypothesized that FHF2A, but not FHF2B, would induce pronounced LTI in the wild-type (WT) Na_v_1.1 channels but to lesser degrees in the F1774S and P1894L mutant channels. While FHF2A did induce strong LTI in WT Na_v_1.1 as expected and FHF2A-induced LTI was significantly attenuated by the P1894L mutation, LTI was induced in the F1774S mutant to a greater extent than that observed with WT channels. This unexpected outcome demonstrates that disease mutations can have a range of effects on the interactions between Nav1.1 and accessory subunits. These variant specific alterations are likely to contribute to distinct disease phenotypes.

## 2. Materials and Methods

### 2.1. SCN1A Channel Constructs

An optimized human cDNA construct was used for WT Nav1.1 [29]. The F1774S and P1894L mutations were introduced into the WT construct using a QuikChange^®^ II XL site-directed mutagenesis kit from Agilent Technologies (Santa Clara, CA, USA) using the manufacturer’s instructions. The mutant channel constructs were fully sequenced by ACGT, Inc. (Chicago, IL, USA) to ensure the correct mutation was present and to check for any additional mutations.

### 2.2. Transfection and Cell Culture

Human Embryonic Kidney cells (HEK293) were transfected to express the WT, F1774S mutant, or P1894L mutant hNav1.1 channels with and without FHF2A or FHF2B. FHF plasmid constructs were obtained from Origene Technologies, Inc. (Rockville, MD, USA). Cells were grown using standard tissue culture conditions. Cells were then transiently transfected using the Invitrogen Lipofectamine 2000 Transfection Reagent following the manufacturer’s instructions or using the calcium phosphate transfection technique (Takara Bio, Inc., San Jose, CA, USA). For the transfection, channel DNA and the appropriate FHF DNA, along with enhanced green fluorescent protein (EGFP) DNA, were mixed with the transfection reagent. The cells were incubated in this mixture for 3 h then incubated in fresh media at 37° C for 24–48 h. After an incubation, whole-cell voltage-clamp recordings were obtained. EGFP, visualized with a fluorescent microscope, was used to identify transfected cells.

### 2.3. Whole-Cell Voltage-Clamp Recordings

Recordings were collected at room temperature with a HEKA (Stuttgart, Germany) EPC-10 amplifier in conjunction with the Pulse (v8.80) and PatchMaster (v2x90.2) programs (HEKA Electronics). Electrodes were made using 1.7 mm capillary glass first with a Sutter P-1000 Micropipette puller (Sutter Instrument Company, Novato, CA, USA), which was then fire-polished to a resistance of 0.8–1.39 MΩ. Recordings were begun 3 min after membrane rupture for each cell to allow the intracellular solution to equilibrate. A series resistance of at least 80% was used to minimize voltage errors. Passive leak currents were also canceled by leak subtraction. All solutions were adjusted to the osmolarity of 300 mOsm with glucose. The extracellular solution was composed of (in mM) 140 NaCl, 3 KCl, 1 CaCl_2_, 1 MgCl_2_, and 10 HEPES. The solution was adjusted to a pH of 7.3 using NaOH. The intracellular solution was composed of (in mM) 140 CsF, 10 NaCl, 1.1 EGTA, and 10 HEPES. The pH of the intracellular solution was adjusted to a pH of 7.3 with CsOH.

Cells with an excessive leak current, abnormal sodium current reversal potentials, or >5 mV uncompensated voltage-clamp errors were not used for the final analysis.

### 2.4. Voltage Protocols and Analysis of Data

#### 2.4.1. Activation Protocol

To measure transient sodium current (I_NaT_), 50 ms depolarizing steps were used from a −100 mV holding potential in 5 mV steps from −80 mV to +65 mV. The current density was found by dividing I_NaT_ by the cell capacitance. The conductance (G_Na_) of the sodium current was measured by conversion of the I_NaT_ using the following equation: G_Na_ = I_NaT_/(V − V_rev_). In this equation, V_rev_ is the Na^+^ reversal potential calculated for each cell by Pulsefit (v8.80) and FitMaster (v90.5) (HEKA Electronics). The average V_rev_ for the 236 cells included in this study was 65.2 ± 0.6 (mean ± standard error). The average series resistance was 2.2 ± 0.1 MΩ, the average series resistance compensation was 81 ± 0.4%, and the average estimated voltage clamp error was 2.9 ± 0.2 mV.

Activation curves were obtained by graphing the normalized G_Na_ versus the depolarizing potentials. Boltzmann functions were then fit using the following equation: G_Na_/G_max_ = 1/(1 + exp [(V_50_, act − V)/k_act_]). In this equation, G_max_ is the peak G_Na_, V_50_ is the half-maximal activation potential, V is the depolarizing potential, and k is the slope of the activation curve. The rate of decay (τNaT) was analyzed in Pulsefit, where current traces were fit using m^3^h Hodgkin and Huxley fits.

#### 2.4.2. Steady-State Inactivation

To measure the availability of the sodium channels, we measured the peak sodium current of a 20 ms test pulse at 0 mV after a 500 ms prepulse at −130 mV to −10 mV in 10 mV increments. The inactivation curves were then obtained by graphing the normalized sodium current versus prepulse potentials. A Boltzmann function was then fit, as previously described [30].

#### 2.4.3. Persistent and Ramp Currents

The persistent sodium current (I_NaP_) was measured at 19–20 ms during a 0 mV depolarizing step and presented as a percentage of the peak current elicited by the 0 mV depolarization. Ramp currents were elicited with a 500 ms ramp depolarization from −100 mV to +40 mV.

#### 2.4.4. Recovery from Inactivation

Channel activation and inactivation were achieved by a 20 ms depolarizing prepulse at 0 mV followed by a repolarizing step to −100 mV for durations from 1 ms to 1025 ms. The non-inactivated currents were then measured at 0 mV during an additional 20 ms test pulse. These were then normalized to the maximum current for that test.

#### 2.4.5. Long-Term Inactivation

Long-term inactivation was determined using a −100 mV holding potential. Cells were held at −100 mV for 60 ms followed by five depolarizing pulses to −20 mV for 16 ms. A −100 mV recovery phase of 40 ms was used between the depolarizing pulses.

#### 2.4.6. Statistics

Statistical analysis and curve fitting were completed using GraphPad Prism (v 9.4.1, GraphPad Software, Boston, MA, USA) and Origin Pro 2025 (OriginLab, Corp., Northampton, MA, USA). Curve fitting used the nonlinear least-squares minimization method. Statistical significance was assessed using one-way ANOVA with Tukey post hoc test, and descriptive statistics were used where appropriate. Data are presented using the mean ± standard error of the mean (SEM) using the presented number of cells.

## 3. Results

To better understand the possible impact of the F1774S and the P1894L mutations on the Nav1.1 structure, we used UCSF Chimera [31] to model these two substitutions. The F1774S mutation was modeled in the Nav1.1 cryo-electron microscopy-derived structure PDB 7dtd [32]. F1774 faces into the pore and is situated roughly halfway between the DEKA selectivity filter and the intracellular narrowing of the pore in the cryo-EM structure (Figure 2).

The serine substitution at position 1774 does not induce any clashes and does not reduce any hydrogen bonds in the structure. However, F1774 may be involved in ligand binding in the pore, and, thus, ligand binding may be altered by the F1774S mutation. It has been hypothesized that the FHF2A LTI particle might bind within the inner pore of the channel [18], and, therefore, we predicted that F1774S could destabilize this interaction and thus reduce LTI. The CTD of Nav1.1 and P1894 are not resolved in the hNav1.1 cryo-EM structure. Therefore, we used the PDB 4jpz X-ray diffraction structure of the Nav1.2 CTD in complex with FHF2B and Ca^2+^/calmodulin [33]. This region of the Nav1.2 CTD is 89% identical with that of Nav1.1, and the residues in the Nav1.2 CTD that interact with the FHF core are 96% identical in Nav1.1, with only one difference among the 28 interacting residues in the structure (E1894 in Nav1.2 corresponds to Q1893 in Nav1.1). P1894 is conserved in Nav1.2 (P1895) and is located at the interface between the FHF core and the Nav1.2 CTD. The P1895L substitution in the structure induces multiple clashes with the FHF core, clashing with FHF2B residues V59 and L51 (Figure 3). Based on this, we predicted that the Nav1.1-P1894L mutation could destabilize the interaction between the FHF core and Nav1.1 and thus indirectly impact the ability of FHF2A to induce LTI by reducing FHF binding to the CTD of Nav1.1.

We expressed WT, F1774S, and P1894L channels in HEK293 cells and used whole-cell patch-clamp electrophysiology to investigate the biophysical properties of the channels. All three constructs expressed robust currents, and representative current traces elicited with step depolarizations are shown in Figure 4A. The voltage dependence of activation was similar for all three channel constructs (Figure 4B); however, the voltage dependence of steady state inactivation was shifted in the depolarizing direction for F1774S channels compared to both WT and P1894L channels (Figure 4C; *p* < 0.001). When analyzing the rate of decay of peak currents associated with fast inactivation (Figure 4D), we found that the P1894L mutant channels are inactivated from the open state more slowly than WT Nav1.1 channels at depolarizations ranging from −25 to 35 mV and more slowly than F1774S channels at depolarizations ranging from −15 to 30 mV. F1774S channels displayed slightly slower inactivation than WT channels at 15 and 20 mV, but, otherwise, the rate of inactivation was not significantly different. Our key questions were (a) can FHFs alter Nav1.1 channel properties and (b) can Nav1.1 disease-associated variants alter the impact of FHFs on Nav1.1 current properties? Therefore, we next co-expressed Nav1.1 and FHF2 constructs in HEK293 cells to investigate the biophysical consequences of FHF2A and FHF2B co-expression and as well as the impact of the two disease variants on Nav1.1 currents. The co-expression of FHF2A or FHF2B did not have any significant impact on the voltage dependence of activation or steady-state inactivation for any of the three channel types (Table 1, Appendix A). The co-expression of FHF2B slightly slowed the rate of inactivation for WT and F1774S channels at several positive test potentials, but FHF2B had no effect on P1894L channels, and the rate of inactivation was not substantially altered for any of the channels co-expressed with FHF2A (Appendix A).

We did not detect a significant difference between WT (n = 31), F1774S (n = 21), and P1894L (n = 25) channels in terms of peak current density, but there were some apparent effects of FHF expression on peak current density (Figure 5A,B; Table 1). The current density of WT channels expressed with FHF2B was significantly greater than that of WT channels expressed alone. The current density of WT channels expressed with FHF2B was also significantly greater than that of P1894L channels (Figure 5B). The co-expression of FHF2A with F1774S channels appears to significantly increase F1774S current density. However, neither FHF2A nor FHF2B impacted P1894L current density.

Previous studies have reported that the F1774S mutation enhances persistent current amplitudes [2,13]. In our experiments, the persistent current, when measured at 0 mV, was found to be increased in F1774S mutant channels (Figure 5C). WT channel persistent currents were not altered by either the FHF2A or FHF2B co-expression. Interestingly, FHF2B co-expression significantly increased F1774S persistent currents, and FHF2A co-expression substantially reduced the F1774S induced persistent currents. This shows a possible rescue of the F1774S phenotype with the addition of FHF2A. The P1894L (Figure 5D) channels showed no increase in the persistent current compared to WT channels and no effect of either FHF2A or FHF2B co-expression.

To further explore the impact of FHF2A and FHF2B on Nav1.1 persistent currents, we used a 500 ms duration ramp depolarization from −100 mV to 40 mV. As can be seen in Figure 6A, F1774S channels generated robust ramp currents at voltages ranging from −50 to +40 mV. WT and P1894L channels produced significantly smaller (*p* < 0.0001) ramp currents. The WT ramp currents observed between −50 and −20 mV likely reflect window currents generated by the overlap between the voltage dependence of activation and steady-state inactivation. Although P1894L channels did not induce significantly larger persistent currents at 0 mV, they appeared to increase the size of the window currents observed between −50 and −10 mV. However, a statistical comparison of the area under the ramp current curves did not find any significant difference. The co-expression of FHF2A and 2B appear to have modest impacts on WT currents induced by the ramp protocol (Figure 6B); however, statistical comparison of the area under the ramp current curves did not find any significant difference between the WT groups. Interestingly, FHF2A co-expression resulted in a substantial reduction in F1774S ramp currents (Figure 6C; *p* < 0.05). Neither FHF2A nor FHF2B had a significant effect on P1894L ramp currents (Figure 6D).

The co-expression of FHF2A led to the substantial long-term inactivation (LTI) of WT Na_v_1.1 channels (Figure 7A). FHF2A also induced robust LTI in F1774S mutant channels (Figure 7B). FHF2B decreased long-term inactivation in both WT and F1774S channels (Figure 7A,B), although the decrease in the LTI of WT channels was not significant when all nine groups were compared with a one-way ANOVA followed by Tukey’s post hoc multiple comparison test (Appendix A). In contrast, with the P1894L mutation located in the FHF binding site, LTI was significantly reduced with the P1894L mutant channels (*p* < 0.0001), and FHF2B did not alter the LTI profile in P1894L channels compared to the control conditions (Figure 7C). Interestingly, the degree of LTI was greater for the F1774S mutant channels when paired with FHF2A than for WT channels (Figure 7D and Figure 8, *p* < 0.05). This significant increase in LTI was unexpected but suggests it may be possible to reduce the over-excitability of some mutant channels by upregulating FHF2A.

Finally, we examined the time course for recovery from inactivation at −100 mV for all nine groups (Figure 9A). All nine groups showed a fast component in their recovery from inactivation (Figure 9B). The fast time constant was smaller for F1774S under all three conditions compared to WT and P1894L channels under all three conditions (control, with FHF2A, and with FHF2B). The fast time constant for WT channels co-expressed with FHF2B was also smaller than that for WT channels under control conditions. The co-expression of FHF2A induced a pronounced slow component in the recovery from inactivation for WT, F1774S and P1894L channels (Figure 9C). The slow time constant was larger for WT channels with FHF2A than for F1774S or P1894L channels with FHF2A. Appendix A shows the recovery from the inactivation time course compared by co-expression condition (Appendix A) and by channel subtype (Appendix A). After 33 ms at −100 mV, fast recovery is mostly complete (>93%; Appendix A). At this time point, fewer F1774S channels with FHF2A co-expression have recovered (54 ± 2%, n = 15; *p* < 0.05) compared to WT channels with FHF2A co-expression (62 ± 1%, n = 23). By contrast, significantly more P1894L channels co-expressed with FHF2A have recovered (80 ± 3%, n = 16; *p* < 0.0001). This is consistent with the enhanced LTI observed with F1774S channels and the reduced LTI observed with P1894L channels when co-expressed with FHF2A.

## 4. Discussion

We investigated if FHF2 variants could modulate the biophysical properties of Nav1.1 voltage-gated sodium channels and two different *SCN1A* variants identified in individuals with distinct disease phenotypes. Our results indicate that FHF2A can significantly impact the gating of wild-type Nav1.1 channels, inducing pronounced long-term inactivation similar to that reported for Nav1.5, Nav1.6, and Nav1.7 channels [18,19,34,35]. Although FHF2B has been reported to modulate the gating properties of other sodium channel isoforms, we observed little impact of FHF2B on Nav1.1 gating in our experiments. However, we did find striking differences in the regulation of two *SCN1A* disease variants by FHF2A. These data suggest that FHFs can be important regulators of Nav1.1 properties and that the interaction between FHF2A and Nav1.1 disease mutations can be differentially impacted by distinct mutations.

Previous studies indicate that FHFs can modulate multiple properties of voltage-gated sodium channels. Both FHF2A and FHF2B have been reported to induce a depolarization of the steady-state inactivation curve of Nav1.5, Nav1.6, and Nav1.7 [19,34,35]. We did not see any change in WT Nav1.1 steady-state inactivation with FHF2A or FHF2B. Our data are consistent with that of Wang et al. [16], who also reported that Nav1.1 steady-state inactivation was not altered by FHF2. We did observe pronounced LTI with Nav1.1 when co-expressed with FHF2A, demonstrating that FHF2A can interact with Nav1.1. Interestingly, Laezza et al. [36] showed that FHF4A induced LTI in Nav1.2 even though neither FHF4A nor FHF4B shifted the steady-state inactivation of Nav1.2. Multiple studies have suggested that FHFs can also modulate the current density of voltage-gated sodium channels. We detected a significant increase in WT Nav1.1 current density with FHF2B current density, but FHF2A did not significantly alter current density. Rush et al. [37] reported that FHF2B, but not FHF2A, similarly enhances Nav1.6 current density. In contrast, Laezza et al. [36] reported that FHF4B substantially decreases the current density of Nav1.2 and Nav1.6 channels. This suggests that investigating the impact of FHF1, FHF3 and/or FHF4 variants on Nav1.1 properties could be informative.

We also observed a significant effect of FHF2A on Nav1.1-F1774S channels. When FHF2A was co-transfected with the F1774S mutant channel, LTI was induced to greater extent than with WT channels. The F1774S mutation substantially increases persistent currents in Nav1.1. The co-expression of FHF2B induced even larger persistent currents at 0 mV. However, the co-expression of FHF2A with F1774S channels greatly reduced persistent currents and ramp currents measured with a slow ramp depolarization. We also found that FHF2A, but not FHF2B, increased the current density of F1774S channels. We speculate that the pronounced decrease in the persistent current paired with the enhanced induction of LTI may lead to at least a partial rescue of the phenotype of the F1774S channel, even with the increased current density.

Mutations to Na_v_1.1 can have complex consequences for gating and electrical activity. Since the channel is found in both excitatory peripheral neurons (dorsal root ganglion neurons, and trigeminal ganglion neurons) and inhibitory interneurons in the central nervous system, it can be hard to predict how particular mutations will alter excitability in the different locations. If specific Nav1.1 mutations have a differential impact on excitability in central and peripheral neurons, it may be difficult to predict the phenotypic changes that will result. Although FHF2 is known to be expressed in both trigeminal ganglia and inhibitory interneurons, it is not yet known which isoforms of the protein are expressed in either location. Multiple FHM3 variants are associated with the enhanced activity of Na_v_1.1 [2,13,38]. Although several FHM3 variants have been reported to have reduced peak current densities [2,38], suggesting a loss of function, neuronal expression experiments and computer modeling suggest that these variants may have an overall gain-of-function impact, in part due to enhanced persistent currents [2,13,39]. Several studies [24,25,40] have suggested that the hyperexcitability of central neurons leads to cortical spreading depression that may underlie the aura associated with migraines. Alternatively, the peripheral neurons in the trigeminal ganglia may play an important role in pain associated with FHM3. Peripheral neurons are a known site of CGRP release, which has been found to induce migraine attacks, and the enhanced activation of Nav1.1 containing sensory neurons can elicit robust pain responses [5]. Regardless of the site of action, the large persistent currents observed with F1774S channels are likely to induce neuronal hyperexcitability. As FHF2A substantially reduces F1774S persistent currents, our data suggest that the upregulation of FHF2A may serve as a useful strategy for attenuating the overexcitability associated with this and other disease-associated variants that exhibit increased persistent current.

The P1894L channel also exhibited gain-of-function changes compared to WT Nav1.1 channels. P1894L slowed the rate of open-channel inactivation. The co-expression of FHF2A with P1894L mutant channels resulted in significantly reduced LTI compared to WT and F1774S channels and, as FHF2A variants that reduce LTI are associated with epilepsy [27], the reduced LTI observed with P1894L can be considered as a gain of function effect. It was predicted that the P1894L mutation, situated in the putative FHF binding site on the CTD, would abolish any FHF binding. Thus, the reduced LTI with the P1894L mutant channels indicates that the FHF2 core likely binds to the CTD of Nav1.1 in a similar fashion to that reported for Nav1.2, effectively increasing the local concentration of the LTI blocking particle around the inner pore of Nav1.1 without impacting steady-state inactivation. FHF2A co-expression still induced some LTI with P1894L channels, indicating that, while the binding of the FHF2 proteins to the CTD may be greatly reduced by the P1894L mutation, the A-tail of FHF2A can still interact with the P1894L channel pore. The P1894L variant was identified in a patient with severe language delay who also had a diagnosis of epilepsy [28]. Many *SCN1A* variants associated with autism spectrum disorder and epilepsy exhibit significant loss-of-function effects [41]. While the P1894L current density tended to be lower than that measured for WT channels, the difference was not significant in our data set. Interestingly, FHFs have also been implicated in regulating sodium channel localization [42,43], and, thus, one interesting possibility is that the P1894L variant alters targeting of the channels to specific neuronal compartments by impacting the binding of FHF proteins to the C-terminal domain of Nav1.1.

Voltage-gated sodium channels are subject to complex regulation by post-translational modifications, accessory subunits, and disease mutations, among other factors. Here, we showed that Nav1.1 current properties can be modified by FHF2A and that specific disease mutations can have differentially altered sensitivities to FHF2A modulation. This raises the possibility that FHFs and voltage-gated sodium channel disease variants can interact in variant specific paradigms that may have important consequences for disease manifestations. Furthermore, it suggests that manipulating FHF expression and/or interaction with voltage-gated sodium channels may have promise for the treatment of disorders of excitability associated with aberrant voltage-gated sodium channel activity.

## Figures and Tables

**Figure 1 cells-14-00291-f001:**
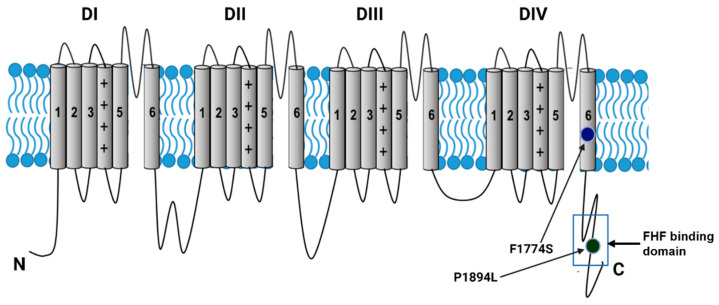
Voltage-gated sodium channel topology with SCN1A mutations investigated and FHF binding domain indicated.

**Figure 2 cells-14-00291-f002:**
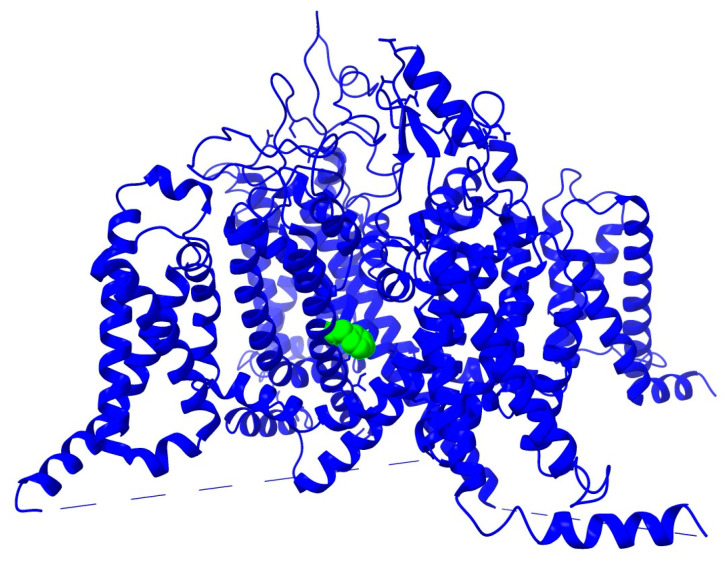
Model of Nav1.1 with the F1774 location highlighted in green. Models are produced using UCSF Chimera version 1.18. The site of the mutation is in the middle of the sixth transmembrane segment and so is part of the pore-forming region of the channel.

**Figure 3 cells-14-00291-f003:**
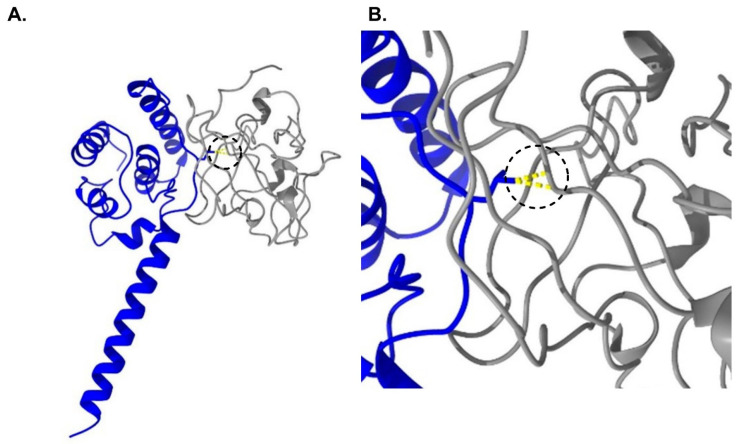
(**A**) The FHF core (gray) binds to the CTD (dark blue) of Nav1.2. The proline residue (P1894 in Nav1.1, P1895 in Nav1.2) is at the interface between the two proteins. (**B**) The model of predicted binding between the CTD of Nav1.1 P1894L mutant channel and the FHF core. Steric clashes induced by the mutation are shown in yellow in the region identified by the dashed circles.

**Figure 4 cells-14-00291-f004:**
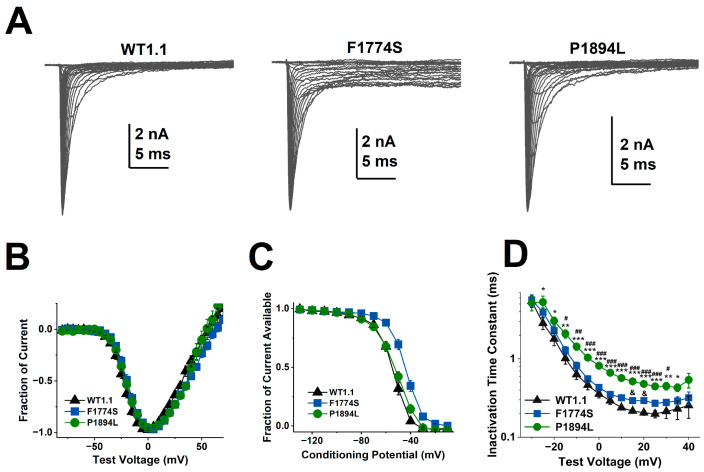
Biophysical properties of WT and mutant Nav1.1 channels transiently expressed in HEK293 cells: (**A**) Representative family of current traces for WT (**left**), F1774S (**middle**) and P1894L (**right**) Nav1.1 channels. (**B**) The normalized current–voltage (*I–V*) properties are assessed using depolarizing steps. Cells are held at −100 mV. The currents are elicited by 50 ms test depolarizations to various potentials from −80 to +65 mV in increments of 5 mV. The peak current evoked by each pulse, normalized to the maximum peak current, is plotted versus the test voltage. There is no difference in the voltage dependence of activation for the WT (black triangles; n = 31), F1774S (blue squares, n = 21), and P1894L (green circles, n = 25). (**C**) The comparison of steady-state inactivation for WT and mutant channels. The F1774S mutation shifts the availability curve in the depolarizing direction (*p* < 0.001). (**D**) P1894L mutant channels have a slower rate of decay compared to the WT. For P1894L versus WT, * *p* < 0.05, ** *p* < 0.001, and *** *p* < 0.0001. For P1894L versus F1774S, # *p* < 0.05, ## *p* < 0.001, and ### *p* < 0.0001. For F1774S versus WT, & *p* < 0.05.

**Figure 5 cells-14-00291-f005:**
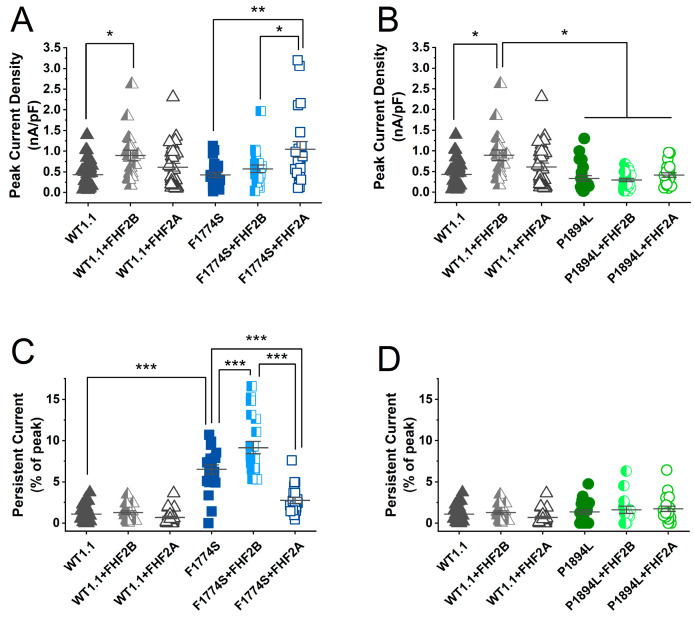
FHF2A has a significant impact on F1774S peak and persistent currents: (**A**) Comparison of the peak current density for WT and F1774S channels measured with a 0 mV step depolarization. (**B**) Comparison of the peak current density for WT and P1894L channels measured with a 0 mV step depolarization. (**C**) Comparison of the relative persistent current amplitude at 0 mV for WT and F1774S channels. (**D**) Comparison of the relative persistent current amplitude at 0 mV for WT and P1894L channels. For all graphs, significant differences are designated as * *p* < 0.05, ** *p* < 0.001, and *** *p* < 0.0001. The mean ± SEM are shown for each group.

**Figure 6 cells-14-00291-f006:**
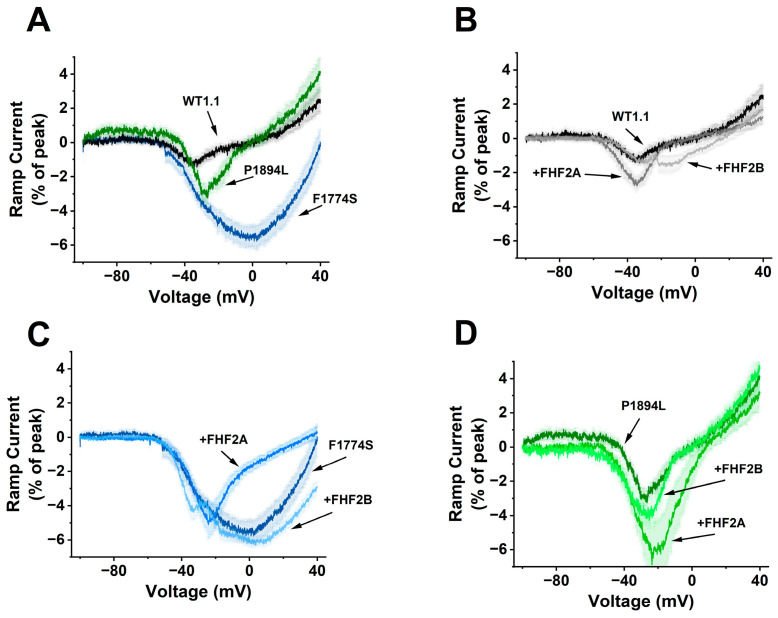
F1774S generates robust ramp currents that are attenuated by FHF2A: (**A**) Comparison of the ramp currents generated by WT, F1774S and P1894L channels without FHF co-expression. (**B**) Comparison of WT ramp currents from WT channels alone and expressed with either FHF2A or FHF2B. (**C**) Comparison of WT ramp currents from F1774S channels alone and expressed with either FHF2A or FHF2B. (**D**) Comparison of WT ramp currents from P1894L channels alone and expressed with either FHF2A or FHF2B. For (**A**–**D**), the ramp current amplitudes are shown as a percentage of the peak transient current elicited with a step depolarization to 0 mV. The thick lines reflect the averaged currents from 14 to 20 cells, and the shaded background for each trace shows the standard error of the means.

**Figure 7 cells-14-00291-f007:**
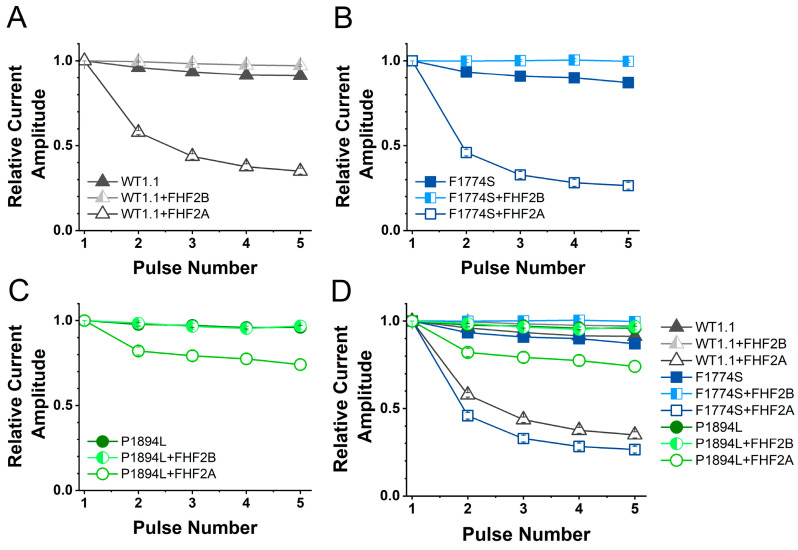
FHF2A induces long-term inactivation in Nav1.1 WT and F1774S mutant channels and, to a lesser extent, with P1894L mutant channels. HEK293 cells transiently expressing human wild-type and mutant Nav1.1 channels with and without the co-transfection of FHF2A or FHF2B and EGFP. Cells are subjected to five depolarizing pulses and normalized to the first pulse. (**A**) The co-transfection of FHF2A-induced LTI in the WT (dark gray, n = 26) compared to the control conditions (black, n = 33) and with FHF2B co-transfection (light gray, n = 21). (**B**) The co-transfection of FHF2A-induced LTI in F1774S channels (blue, n = 20) while the co-transfection of FHF2B significantly reduces the baseline current attenuation of F1774S (light blue, n = 22) channels compared to control F1774S (dark blue, n = 23). (**C**) The co-transfection of FHF2A induces LTI to a lesser extent in P1894L (green, n = 22) mutant channels (*p* < 0.0001). The co-transfection of FHF2B does not alter LTI P1984L channels (light green, n = 22) compared to the control P1984L (dark green, n = 26). (**D**) The LTI data are compared for all three channels under the three conditions. Error bars indicate SEM.

**Figure 8 cells-14-00291-f008:**
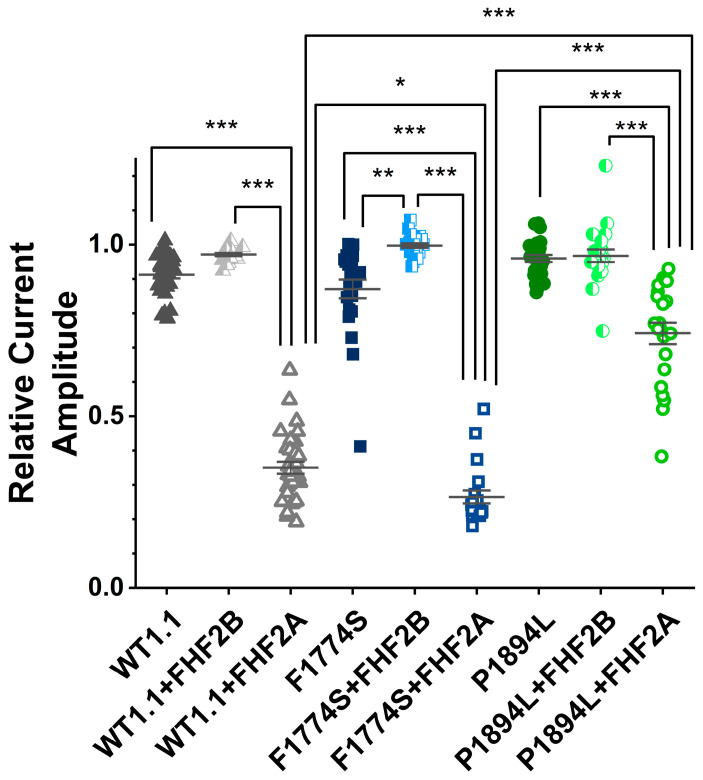
The final pulse of the long-term inactivation protocol shows differences in LTI. HEK293 cells transiently expressing human wild-type and mutant Nav1.1 channels with and without the co-transfection of FHF2A and FHF2B proteins and EGFP. As above, the addition of FHF2A, but not FHF2B, leads to accumulation in a long-term inactivated state for WT channels, enhanced LTI for F1774S channels, and significantly reduced LTI with P1894L channels. Significant results of interest are shown for key comparisons. * *p* < 0.05; ** *p* < 0.001; *** *p* < 0.0001. The mean ± SEM are shown for each group.

**Figure 9 cells-14-00291-f009:**
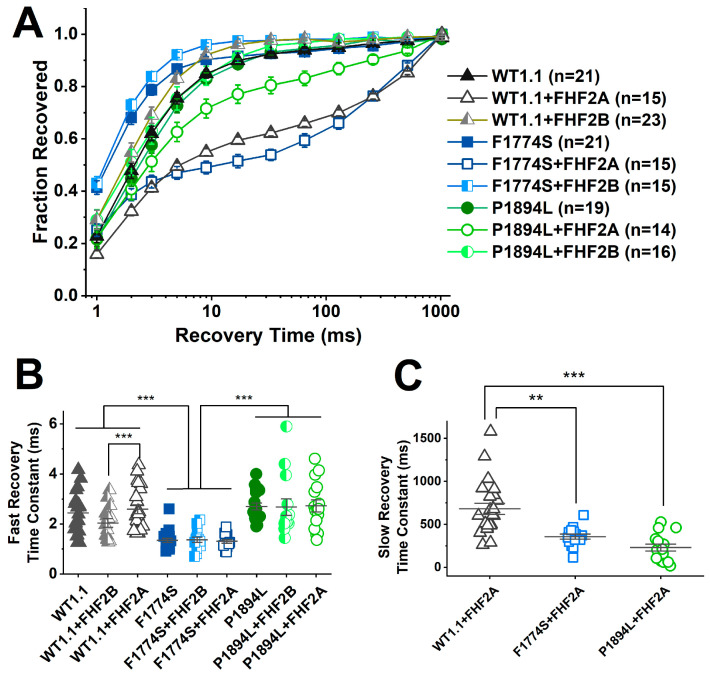
The F1774S mutation and FHF2A differentially impact recovery from inactivation: (**A**) Recovery from inactivation time course is shown for HEK293 cells transiently expressing human WT and mutant Nav1.1 channels with and without the co-transfection of FHF2A and FHF2B proteins. FHF2A co-expression induces a slow recovery component in WT (gray open triangles) and F1774S (blue open squares) channels and, to a lesser extent, in P1894L (green open circles) channels. (**B**) The fast time constants of recovery are shown for all nine conditions. The fast time constant is smaller for F1774S channels with and without FHFs compared to the other six groups. (**C**) The slow time constant is compared for the three groups co-expressing FHF2A. The slow time constant is larger for WT channels with FHF2A than for either F1774S or P1894L channels with FHF2A. The number of cells in each group are indicated in (**A**). Significant results are indicated as follows: ** *p* < 0.001; and *** *p* < 0.0001. The means ± SEM are shown.

**Table 1 cells-14-00291-t001:** The V_1/2_ and slope of the voltage dependence of activation (G/GMax) and the steady-state inactivation (I/Imax) as well as the current density and fast inactivation (τ) are shown. Significance is indicated as follows: * *p* < 0.05 compared to the WT; # *p* < 0.05 compared to FS. The mean ± SEM are provided for each measure.

**Constructs**	Activation		Inactivation		Peak Current Density	Decay	
	V_1/2_ (mV)	κ	V_1/2_ (mV)	κ	(pA/pF)	τNaT at 0 mV (ms)	n
WT1.1	−21.6 ± 1.4	6.8 ± 0.2	−53.3 ±1.1	−5.3 ± 0.1	431 ± 57	0.36 ± 0.04	31
WT1.1+FHF2A	−22.2 ± 1.9	7.7 ± 0.3	−52.5 ±1.7	−6.2 ± 0.4	613 ± 93	0.47 ± 0.04	32
WT1.1+FHF2B	−23.5 ± 2.1	6.0 ± 0.2	−52.4 ± 1.6	−4.9 ± 0.1	892 ± 130 *	0.60 ± 0.07	20
F1774S	−16.1 ± 1.4	6.8 ± 0.3	−45.0 ± 1.3 *	−5.5 ± 0.2	427 ± 68	0.43 ± 0.03	21
F1774S+FHF2A	−16.2 ± 2.1	8.0 ± 0.7	−42.0 ± 1.7 *	−4.3 ± 0.2	1042 ± 193 *, #	0.56 ± 0.08	20
F1774S+FHF2B	−19.4 ± 2.6	6.4 ± 0.3	−43.2 ± 1.2 *	−5.4 ± 0.2	574 ± 91	0.73 ± 0.04	20
P1894L	−19.6 ± 2.0	7.2 ± 0.2	−52.6 ± 1.5 #	−7.1 ± 0.5 *, #	336 ± 66	0.82 ± 0.06 *, #	25
P1894L+FHF2A	−19.8 ± 1.2	6.8 ± 0.3	−48.6 ± 2.1	−7.3 ± 0.3 *, #	419 ± 53	1.13 ± 0.16 *, #	23
P1894L+FHF2B	−22.0 ± 2.5	6.8 ± 0.2	−48.9 ± 2.5	−6.1 ± 0.2	298 ± 40	1.07 ± 0.10 *, #	23

## Data Availability

The original contributions presented in this study are included in this article. The raw data supporting the conclusions of this article will be made available by the authors upon request.

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
