# Peer review of "Differential Regulation of Nav1.1 and SCN1A Disease Mutant Sodium Current Properties by Fibroblast Growth Factor Homologous Factors"

_cells, 2025, doi:10.3390/cells14040291_

Round 1
Reviewer 1 Report
Comments and Suggestions for Authors
The manuscript under review from Frazee et al. describes how the interaction with FHF2 molecules affect the biophysical properties of WT Nav1.1 channel as well as of two disease associated mutations, F1774L and P1894L, causing familial hemiplegic migraine 3 and epilepsy respectively,.
The manuscript is not always correctly written, sentences aresometimes unclear or poorly structured. Methods are suited
for the objectives. Description of the results as well as graphic
rendering of figures have to be improved . Some figures even
contain errors. For all these reasons, I consider this manuscript not suitable
for publication in the current form, even if I think that after a
deep and substantial correction it can improve a lot and be
considered for publication
Below I list the major and minor revisions to be made,
but at the same time I suggest that the authors thoroughly
check the entire manuscript in every part. Above all the introduction needs to be arranged more logically. Major revisions As a general premise I think it is necessary to extend statistics
to the data reported in the figures and table when the authors
want to attribute relevance to an observed difference.
This can help to distinguish between important results to discuss and slight differences.
Lines 34-36 The sentence can generate confusion. Nav1.1 is expressed in both excitatory and inhibitory neurons of CNS, but its physiological relevance is higher in GABAergic inhibitory neurons.
Nav1.1 expression in peripheral nervous system is present as well (for instance in the dorsal root ganglion), but its role and its functional prevalence has not well defined yet . At least as far as I know.
Line 64 Long-term inactivation (LTI) has to be introduced and briefly explained
Figure 4 is difficult to read; symbols should be larger and somehow better distinguishable from each other , also in the legend. y-axis ticks can be reduced in number.
Figure 5 I don’t see the statistic between F1774S and F1774S+ FHF2B. Moreover it should be indicated the values of significance corresponding to the various number of asterisks.
Figure 6 Current density values seem really huge. I think someerror may have occurred. It is not easy to handle such big currents
with patch clamp. Moreover it is important to known whether the
observed differences are statistically relevant or not. If yes
this confers greater relevance to the data. X axis is not uniform for
A,B and C,D. As in figure 4 symbols should be larger and more
distinguishable. Line 296 and following , Table 1 . WT values of the voltage of
steady state activation (-39 mV) is significantly more negative
than that generally found (-20,-25 mV). Can you explain this
discrepancy ? Figure 8 is not very informative and, most importantly,
plots disagree with the values of the tau of recovery
from inactivation of Table 1. There is a big difference
in tauNar value between WT (19.83) and P1894L (41.01)
that is not detectable in Figure 8. Discussion. Some conclusions included in the discussion
do not seem supported by the data shown in Table 1,
for which the statistical significance has not been calculated.
For example, sentences reported in Lines 406-409 and 433-437.
My suggestion is that the authors review and rewrite the discussion after making the necessary corrections in the results section and after calculating the statistical significances of the data presented
Minor revisions
Lines 44-46 Past studies have shown this mutation to cause increased persistent current (INaP) in HEK-293 ….
Please modify as follow : Past studies have shown that this mutation causes an increase…
Lines 57-57 Here we first asked if fibroblast growth factor homologous factors (FHFs) could
modulate Nav1.1 properties as well as Nav1.1-F1774S channels
This sentence has to be moved later in the introduction where the purposes of the paper are described
Line 75-76 to involve a proline at amino acid 149 in FHF1B (FGF12B) and the corresponding location P154
Please modify as follow: a proline in position 149 (P149)
Line 80 For PNS excitatory etc see above (lines 34-36)
Line 90 To further interrogate the interaction between FHF2….
Please modify as follow: to further investigate
Line 99 Why did you make in advance this hypothesis?
Line 110 Could you please briefly describe how this construct has been optimized?
Are FHF plasmid constructs too? If yes, this should be mentioned
Line 132 1.0-1.39
Are you sure of these numbers?
Line 134 series resistances..
Do you mean that only cells where at least 80% of series resistance could be corrected were accepted ? Or what? Please correct
Line 244, 318 and others. I wouldn’t say “treatment condition” but only “conditions”
Line 270 and others. Idem as above instead of “treatment” write “condition”
Comments on the Quality of English LanguageThe manuscript is not always correctly written, sentences are sometimes unclear or poorly structured. I think that the paper needs moderate English revision.
Author Response
We thank the reviewers for their time and their constructive comments. We did a reanalysis of the original data set and discovered an issue with the voltage-offset that impacted about 75% of the original cells. We therefore did multiple transfected and recorded from over 200 new cells. The major findings of the original study were confirmed, but there are some differences. All of the data figures have been completely redone and the manuscript has been rewritten and reorganized to improve it. We also added additional supplemental figures and tables. We note that the major findings (that FHF2A can modulate Nav1.1 LTI, that disease variants can differentially impact LTI of Nav1.1 induced by FHF2A and that FHF2A can substantially reduce persistent currents induced by the F1774S mutation) were confirmed in the new dataset. We are confident in the data we are presenting and our major findings.
Below we provide a point-by-point response to the prior review.
Reviewer #1:
The manuscript is not always correctly written, sentences are
sometimes unclear or poorly structured. Methods are suited
for the objectives. Description of the results as well as graphic
rendering of figures have to be improved . Some figures even
contain errors. For all these reasons, I consider this manuscript not suitable
for publication in the current form, even if I think that after a
deep and substantial correction it can improve a lot and be
considered for publication
Response: We have collected new data, revised most figures and improved the writing as suggested.
Below I list the major and minor revisions to be made,
but at the same time I suggest that the authors thoroughly
check the entire manuscript in every part. Above all the introduction needs to be arranged more logically.
Response: We have checked every part of the manuscript. We believe that the introduction is better organized now.
Major revisions As a general premise I think it is necessary to extend statistics
to the data reported in the figures and table when the authors
want to attribute relevance to an observed difference.
This can help to distinguish between important results to discuss and slight differences.
Response: We have revised the figures so that significant findings are noted in the text and figures. With nine experimental groups, this can get potentially confusing, therefore we have tried to focus on the most important differences and designed the figures to help highlight the most impactful findings.
Lines 34-36 The sentence can generate confusion. Nav1.1 is expressed in both excitatory and inhibitory neurons of CNS, but its physiological relevance is higher in GABAergic inhibitory neurons.
Response: We agree, revised the sentence and included references for context. (lines 34-39)
Nav1.1 expression in peripheral nervous system is present as well (for instance in the dorsal root ganglion), but its role and its functional prevalence has not well defined yet . At least as far as I know.
Response: We agree that the role of Nav1.1 in the peripheral nervous system has been understudied. Osteen et al. (now cited) reported that Nav1.1 can play a significant role in pain and may play an important role in migraine. The revised manuscript reflects the current state of knowledge.
Line 64 Long-term inactivation (LTI) has to be introduced and briefly explained
Response: Thanks for the suggestion. This is now included on lines 66-71.
Figure 4 is difficult to read; symbols should be larger and somehow better distinguishable from each other , also in the legend. y-axis ticks can be reduced in number.
Response: This figure and others have been revised to make it easier to visualize the data and important differences.
Figure 5 I don’t see the statistic between F1774S and F1774S+ FHF2B. Moreover it should be indicated the values of significance corresponding to the various number of asterisks.
Response: This figure and others have been revised to make it easier to visualize the data and statistically significant differences.
Figure 6 Current density values seem really huge. I think some
error may have occurred. It is not easy to handle such big currents
with patch clamp. Moreover it is important to known whether the
observed differences are statistically relevant or not. If yes
this confers greater relevance to the data. X axis is not uniform for
A,B and C,D. As in figure 4 symbols should be larger and more
distinguishable. Line 296 and following , Table 1 . WT values of the voltage of
steady state activation (-39 mV) is significantly more negative
than that generally found (-20,-25 mV). Can you explain this
discrepancy ? Figure 8 is not very informative and, most importantly,
plots disagree with the values of the tau of recovery
from inactivation of Table 1. There is a big difference
in tauNar value between WT (19.83) and P1894L (41.01)
that is not detectable in Figure 8. Discussion. Some conclusions included in the discussion
do not seem supported by the data shown in Table 1,
for which the statistical significance has not been calculated.
For example, sentences reported in Lines 406-409 and 433-437.
Response: As noted, we discovered a V-offset error that impacted about two-thirds of the original recordings. Therefore we collected data from over 200 new cells (roughly 20+ cells per group). We use large pipette tips, so we are able to clamp fairly large sodium currents with low voltage-clamp errors. Cells that had clear voltage-clamp errors or large theoretical voltage clamp errors were excluded from most of the analysis. The average theoretical uncompensated voltage-clamp error was less than 3 mV. As can be seen in figure 4 and supplemental figure S1, we have good current voltage curves. The recovery data is all new.
My suggestion is that the authors review and rewrite the discussion after making the necessary corrections in the results section and after calculating the statistical significances of the data presented
Response: The discussion has been revised to reflect the statistically significant results.
Minor revisions
Lines 44-46 Past studies have shown this mutation to cause increased persistent current (INaP) in HEK-293 ….
Please modify as follow : Past studies have shown that this mutation causes an increase…
Response: Thank you for the careful reading and suggestion. We have revised the sentence (lines 46-49).
Lines 57-57 Here we first asked if fibroblast growth factor homologous factors (FHFs) could
modulate Nav1.1 properties as well as Nav1.1-F1774S channels
This sentence has to be moved later in the introduction where the purposes of the paper are described
Response: Thank you for the careful reading and suggestion. We have revised the introduction and believe that the organisation is appropriate.
Line 75-76 to involve a proline at amino acid 149 in FHF1B (FGF12B) and the corresponding location P154
Please modify as follow: a proline in position 149 (P149)
Response: Thank you for the careful reading and suggestion. We have revised the sentence (line 80)
Line 80 For PNS excitatory etc see above (lines 34-36)
Response: Osteen et al. reported that Nav1.1 activation can significantly increase the excitability of peripheral neurons and thus can induce pain related behaviours. Although the role of Nav1.1 in the peripheral nervous system is not well understood, this may be because it has been understudied. Osteen et al. postulated in their Nature paper that Nav1.1 in trigeminal neurons may play a role in migraine and other pain syndromes. We believe that the potential impact of FHM3 variants on peripheral neuron excitability is unclear but should not be ignored.
Line 90 To further interrogate the interaction between FHF2….
Please modify as follow: to further investigate
Response: investigate and interrogate are synonyms.
Line 99 Why did you make in advance this hypothesis?
Response: Our rationale for the hypothesis is included in lines 202-207 in the revised manuscript: F1774 may be involved in ligand binding in the pore, and thus ligand binding may be altered by the F1774S mutation. It has been hypothesized that the FHF2A LTI particle might bind within the inner pore of the channel [18], and therefore we predicted that F1774S could destabilize this interaction and thus reduce LTI.
Line 110 Could you please briefly describe how this construct has been optimized?
Are FHF plasmid constructs too? If yes, this should be mentioned
Response details about the optimized Nav1.1 construct are found in reference 29. The FHF plasmids were purchased from Origene and this is now mentioned in the methods.
Line 132 1.0-1.39
Are you sure of these numbers?
Yes, we use big tips to reduce the voltage-clamp error.
Line 134 series resistances..
Do you mean that only cells where at least 80% of series resistance could be corrected were accepted ? Or what? Please correct
We have clarified the average series resistance compensation used and the theoretical maximum voltage-clamp error in the methods (lines 153-155).
Line 244, 318 and others. I wouldn’t say “treatment condition” but only “conditions”
Response: Agreed. Changes have been made as suggested.
Line 270 and others. Idem as above instead of “treatment” write “condition”
Comments on the Quality of English Language
The manuscript is not always correctly written, sentences are sometimes unclear or poorly structured. I think that the paper needs moderate English revision.
Response: We have carefully edited the manuscript to improve the English.
Reviewer 2 Report
Comments and Suggestions for Authors
The manuscript of Frazee et al. entitled:” Differential regulation of Nav1.1 and SCN1A disease mutant sodium current properties by fibroblast growth factor homologous factors” explores the functional effects of two isoforms of fibroblast factor homologous factors (FHFs), FHF2A and FHF2B, on human sodium channel (Nav1.1) activity. Additionally, the authors investigated not only the regulation of the wild-type Nav1.1 channel but also examined how specific point mutations in this channel, associated with a pathological neurological phenotype, might influence the functional impact of the FHF isoforms. The manuscript presents a highly interesting topic, particularly in how the authors have explored the molecular mechanisms underlying the functional interaction between the Nav1.1 channel and FHF isoforms. However, the clarity of the authors' message could be improved, and the interpretation and arguments should be sharpened. A list of major and minor issues that need to be addressed is provided below.
Major issues:
1. Fig. 4: To clearly show whether there are statistically significant or non-significant differences between the long-term inactivation of wild-type and mutant Nav1.1 channels, with or without FHF isoforms, data from pulse 6 should be presented in bar graphs.
2. Fig 6C: I have a concern about statistical significance of the functional effect of FHF2A on the P1894L mutant compared to the P1894L mutant alone. It would be helpful to provide statistical analysis for the −20 mV data. The same consideration applies to Figure 6D. If no significant difference is found, it may call into question the conclusion that the P1894L mutant still interacts with the FHF2A and 2B. This should be carefully addressed by the authors. An immunoprecipitation experiment could provide a more direct answer to this question. It is just suggestion for the future experiments.
3. Line 19-20: The authors suggested that co-expressing FHF2A with the F1774S mutant might potentially rescue the pathological mutant phenotype. However, there remains uncertainty about whether the interaction between the Nav1.1 channel and FHF2A is sustained in neurons or if it is only temporary. If the interaction is persistent, effect of FHF2A might be too pronounced, especially concerning long-term inactivation and current density, although it does balance the fraction of cells recovered from inactivation and the absolute persistent current at -10 mV. The key question is which of these parameters is responsible for the pathological phenotype. Without this knowledge, drawing conclusions about the therapeutic potential of the interaction between the Nav1.1 mutant and FHF2A is challenging.
Minor issues:
1. Line 50: The authors noted that several FHM3 mutations have been identified with significant gain-of-function effects. It would be helpful to mention some of these mutations specifically.
2. It would be beneficial for readers if representative current traces were included in Figures 4 and 5.
Comments on the Quality of English LanguageThe manuscript could benefit from a bit of English polishing to improve clarity and readability. I suggest making minor revisions to the language to ensure that the authors' ideas are communicated as effectively as possible.
Author Response
We thank the reviewers for their time and their constructive comments. We did a reanalysis of the original data set and discovered an issue with the voltage-offset that impacted about 75% of the original cells. We therefore did multiple transfected and recorded from over 200 new cells. The major findings of the original study were confirmed, but there are some differences. All of the data figures have been completely redone and the manuscript has been rewritten and reorganized to improve it. We also added additional supplemental figures and tables. We note that the major findings (that FHF2A can modulate Nav1.1 LTI, that disease variants can differentially impact LTI of Nav1.1 induced by FHF2A and that FHF2A can substantially reduce persistent currents induced by the F1774S mutation) were confirmed in the new dataset. We are confident in the data we are presenting and our major findings.
Below we provide a point-by-point response to the prior review.
Reviewer 2
Comments and Suggestions for Authors
The manuscript of Frazee et al. entitled:” Differential regulation of Nav1.1 and SCN1A disease mutant sodium current properties by fibroblast growth factor homologous factors” explores the functional effects of two isoforms of fibroblast factor homologous factors (FHFs), FHF2A and FHF2B, on human sodium channel (Nav1.1) activity. Additionally, the authors investigated not only the regulation of the wild-type Nav1.1 channel but also examined how specific point mutations in this channel, associated with a pathological neurological phenotype, might influence the functional impact of the FHF isoforms. The manuscript presents a highly interesting topic, particularly in how the authors have explored the molecular mechanisms underlying the functional interaction between the Nav1.1 channel and FHF isoforms. However, the clarity of the authors' message could be improved, and the interpretation and arguments should be sharpened. A list of major and minor issues that need to be addressed is provided below.
Response: We thank the reviewer for their comments and we have done extensive work to improve the manuscript.
Major issues:
- 4: To clearly show whether there are statistically significant or non-significant differences between the long-term inactivation of wild-type and mutant Nav1.1 channels, with or without FHF isoforms, data from pulse 6 should be presented in bar graphs.
Response: We have included the data on the final pulse of the LTI protocol in Figure 8 with statistics. We also provide the full Anova and Tukey post-hoc test results for these data in supplemental Tables 1 and 2.
- Fig 6C: I have a concern about statistical significance of the functional effect of FHF2A on the P1894L mutant compared to the P1894L mutant alone. It would be helpful to provide statistical analysis for the −20 mV data. The same consideration applies to Figure 6D. If no significant difference is found, it may call into question the conclusion that the P1894L mutant still interacts with the FHF2A and 2B. This should be carefully addressed by the authors. An immunoprecipitation experiment could provide a more direct answer to this question. It is just suggestion for the future experiments.
Response: We have redone all of the experiments and in our revised data set we do not see a significant difference in the current density between PL expressed with or without FHF2A. We do see some LTI when PL is expressed with FHF2A. This could reflect free floating FHF2A interacting with the pore of PL. Immunoprecipitation might help resolve this. We hope in future experiments to investigate this more fully.
- Line 19-20: The authors suggested that co-expressing FHF2A with the F1774S mutant might potentially rescue the pathological mutant phenotype. However, there remains uncertainty about whether the interaction between the Nav1.1 channel and FHF2A is sustained in neurons or if it is only temporary. If the interaction is persistent, effect of FHF2A might be too pronounced, especially concerning long-term inactivation and current density, although it does balance the fraction of cells recovered from inactivation and the absolute persistent current at -10 mV. The key question is which of these parameters is responsible for the pathological phenotype. Without this knowledge, drawing conclusions about the therapeutic potential of the interaction between the Nav1.1 mutant and FHF2A is challenging.
Response: We agree that FHF2A might not rescue the clinical phenotype. We have clarified that we think FHF2A expression might “lead to a possible rescue of the mutant channel phenotype.” We have added new data on ramp currents to further show that FHF2A can reduce the aberrant currents induced by F1774S. There is quite a bit of evidence indicating that persistent currents are key drivers of neuronal excitability and likely to be important contributors to pathological phenotypes.
Minor issues:
- Line 50: The authors noted that several FHM3 mutations have been identified with significant gain-of-function effects. It would be helpful to mention some of these mutations specifically.
Response: We have increased our refences of FHM3 studies that have identified gain-of-function effects and noted that in at least one study loss-of-function was identified with some FHM3 variants. We did not go into detail about where the various variants are located in order to keep the introduction and discussion concise.
- It would be beneficial for readers if representative current traces were included in Figures 4 and 5.
Response: We include representative traces in the revised figure 4.
Comments on the Quality of English Language
The manuscript could benefit from a bit of English polishing to improve clarity and readability. I suggest making minor revisions to the language to ensure that the authors' ideas are communicated as effectively as possible.
Response: We have carefully edited the manuscript to improve clarity.
Round 2
Reviewer 1 Report
Comments and Suggestions for Authors
The paper has been extensively reviewed and has been much improved in comparison to the previous version. I thank the authors since they decided to satisfy almost all my requests.
Two further observations:
1) Legend of figure 4 . There are some redundancies in the legend of figure 4. The authors have to rearrange properly A and B section of the legend .
2) Discussion: lane 451. The divergence discussed in reference 38 was overcome by the results found later (see ref 2-13). The authors have to modify the sentence.
Author Response
We thank Reviewer #1 for their time and their constructive comments.
Below we provide a point-by-point response to the prior review.
Two further observations:
- Legend of figure 4 . There are some redundancies in the legend of figure 4. The authors have to rearrange properly A and B section of the legend.
Response: Thank you for the careful reading of the legend. We agree and we have edited the legend and removed the redundancies.
- Discussion: lane 451. The divergence discussed in reference 38 was overcome by the results found later (see ref 2-13). The authors have to modify the sentence.
Response: Thank you for your feedback. We agree and we have revised the sentence to reflect that while some variants can reduce peak current density, they seem to have overall gain-of-function effects, in part due to persistent currents. We cite the appropriate papers in the revised sentence, including a new reference ([39]).
Reviewer 2 Report
Comments and Suggestions for Authors
The authors have addressed all my questions. They have conducted a solid study that offers valuable insights into the topic. The revised manuscript meets the standards for publication and will make a meaningful contribution to the field.
Author Response
We thank Reviewer #2 for their time and their positive feedback.